# Phylogeny and Genomic Characterization of Clinical *Salmonella enterica* Serovar Newport Collected in Tennessee

Lauren K. Hudson,[a] William E. Andershock,[b] Xiaorong Qian,[c] Paula L. Gibbs,[c] Kelly Orejuela,[d] Katie N. Garman,[d] John R. Dunn,[d] Thomas G. Denes[a]

[a]Department of Food Science, University of Tennessee, Knoxville, Tennessee, USA
[b]Department of Public Health, University of Tennessee, Knoxville, Tennessee, USA
[c]Division of Laboratory Services, Tennessee Department of Health, Nashville, Tennessee, USA
[d]Tennessee Department of Health, Nashville, Tennessee, USA

**ABSTRACT** *Salmonella enterica* subsp. *enterica* serovar Newport (*S.* Newport) is a clinically and epidemiologically significant serovar in the United States. It is the second most prevalent clinically isolated *Salmonella* serovar in the United States, and it can contaminate a wide variety of food products. In this study, we evaluated the population structure of *S.* Newport clinical isolates obtained by the Tennessee Department of Health during routine surveillance (*n* = 346), along with a diverse set of other global clinical isolates obtained from EnteroBase (*n* = 271). Most of these clinical isolates belonged to established lineages II and III. Additionally, we performed lineage-specific phylogenetic analyses and were able to identify 18 potential epidemiological clusters among the isolates from Tennessee, which represented a greater proportion of Tennessee isolates belonging to putative epidemiological clusters than the proportion of isolates of this serovar that are outbreak related.

**IMPORTANCE** This study provides insight on the genomic diversity of one of the *Salmonella* serovars that most frequently cause human illness. Specifically, we explored the diversity of human clinical isolates from a localized region (Tennessee) and compared this level of diversity with the global context. Additionally, we showed that a greater proportion of isolates were associated with potential epidemiological clusters (based on genomic relatedness) than historical estimates. We also identified that one potential cluster was predicted to be multidrug resistant. Taken together, these findings provide insight on *Salmonella enterica* serovar Newport that can impact public health surveillance and responses and serve as a foundational context for the *Salmonella* research community.

**KEYWORDS** *Salmonella*, *Salmonella* Newport, whole-genome sequencing

*Salmonella enterica* subsp. *enterica* serovar Newport (*S.* Newport) is a clinically and epidemiologically significant serovar in the United States. From the most recent National *Salmonella* Surveillance Overview report in 2016, *S.* Newport was the second most prevalent serovar in culture-confirmed human *Salmonella* infections reported in the Laboratory-based Enteric Disease Surveillance (LEDS) system, accounting for 10.1% of the human *Salmonella* infections reported to LEDS that year (1). This represented a 40.1% increase in reported *S.* Newport infections from 2006 to 2016 (1). Additionally, *S.* Newport is the third most common *Salmonella enterica* serovar identified among the estimated 1.2 million human salmonellosis infections that occur annually in the United States (1). Overall, nontyphoidal *Salmonella* infections are responsible for an estimated 23,000 hospitalizations and 450 deaths per year in the United States (2). Of a total of 4,779 *S.* Newport cases identified by FoodNet from 1996 to 2006, 21.9% resulted in hospitalization, 1.4% in invasive disease, and 0.3% in death (3).

S. Newport is commonly found in the southeastern region of the United States. According to the LEDS report, Tennessee has an incidence rate (IR) for *S.* Newport of 1.44 per 100,000 population (1), which is above average compared to other states. The states with the highest

Address correspondence to Thomas G. Denes, tdenes@utk.edu.

The authors declare no conflict of interest.

IR (per 100,000 population) of culture-confirmed human *S.* Newport reported to LEDS in 2016 were South Carolina (7.45), Mississippi (6.72), Louisiana (6.10), and Arkansas (5.31). Incidence for *S.* Newport is highest in late summer, specifically in the month of August (2). From January through April, incidence for *S.* Newport is low, but from May through November the incidence is higher than baseline (2).

*Salmonella* serovars vary in ability to infect or colonize different hosts and thrive in specific environments. Typically, *S.* Newport is a predominant human serovar in the United States and can be transmitted to humans from consumption of a wide variety of contaminated foods, such as beef, pork, poultry, milk, and produce (4, 5). *S.* Newport is recognized as one of the major serovars in cattle (6) and is capable of long-term survival in manure environments, which can serve as reservoirs for animal and human infection (4). Additionally, wild animals and associated natural environments can provide ecological reservoirs for *S.* Newport (7, 8). Additionally, *S.* Newport can invade plant tissues, which renders washing and disinfection ineffective for eliminating this pathogen during processing (4, 9). An outbreak of *S.* Newport occurred in 2020 where 1,127 people across 48 states were infected by contaminated onions (5). Overall, *S.* Newport is a pathogen that can survive in a variety of environments, which enables this bacterium to utilize several different vectors to infect a host.

*S.* Newport is a confirmed polyphyletic serovar consisting of three genetic lineages, Newport-I, Newport-II, and Newport-III (10). Newport-I is typically found in Europe and frequently isolated from human sources (10). Newport-II and Newport-III are predominantly found in North America, are isolated from human sources at slightly lower frequencies than Newport-I, and contain isolates from nonhuman mammals and reptiles. All three lineages can readily infect humans, but Newport-I preferentially infects humans instead of reptiles or other nonhuman mammals (10). However, Newport-I is rarer than the other two lineages, and many human isolates are from either Newport-II or -III (10).

Based on the U.S. Centers for Disease Control and Prevention *Antibiotic Resistance Threats in the United States, 2019* report, drug-resistant nontyphoidal *Salmonella* is a serious threat, with 212,500 estimated drug-resistant infections per year resulting in an estimated $400 million in direct medical costs (11). Typically, antibiotics are not used to treat *Salmonella* cases, but when their use is needed, ampicillin (beta-lactam), chloramphenicol (amphenicol), ciprofloxacin (quinolone), ceftriaxone (beta-lactam), trimethoprim-sulfamethoxazole (folate pathway antagonist, sulfonamide), amoxicillin (beta-lactam), and azithromycin (macrolide) are used (12–16).

Salmonellosis caused by *S.* Newport is a major public health concern; it is a clinically significant serovar in the southeast and is prevalent in the state of Tennessee. The objectives of this study were to (i) determine the population structure of clinical *S.* Newport in Tennessee to provide insights on the proportion of disease-associated genotypes within the state, compared to global diversity; (ii) identify putative epidemiological clusters based on a single-nucleotide polymorphism (SNP) distance threshold to estimate the proportion of cases that are sporadic or cluster-associated; and (iii) identify antibiotic resistance (ABR) determinants to predict resistance and compare our findings to national and global levels.

## RESULTS AND DISCUSSION

This analysis included a set of 346 *Salmonella* serovar Newport clinical isolates (Table 1; see also Data Set S1 in the supplemental material) sequenced by the Tennessee Department of Health as part of routine surveillance, with collection dates from 2017 to 2021. On average, the Tennessee isolate assemblies consisted of 48.1 contigs, 4.755 Mb in length, with 52.15% G+C content (Data Set S1). The assembly lengths and G+C contents were within expected ranges for this serovar (17, 18). Additional clinical isolates were obtained from EnteroBase ($n = 271$) (Table 1; Data Set S1) and were included in the analyses. These isolates represented 11 different countries and were selected to represent the global diversity of clinical *S.* Newport and provide global context for the isolates collected in Tennessee.

**Clinical *Salmonella* serovar Newport populations consist of three genetic lineages.** Previous studies had established that this serovar contains a high level of genomic diversity and is polyphyletic, with three lineages (10, 18, 19), and this was consistent with our findings

**TABLE 1** Number of isolates by source, isolation location, lineage, and ceBG[a]

| Source | Location | All isolates | No. of isolates with indicated lineage and ceBG (HC900) | | | | | | | | | | | | |
| | | | I | | | | II | | | | | III | | |
| | | | All | 370 | 12161 | 135213 | All | 44 | 391 | 401 | Unknown | All | 120 | 65092 |
|---|---|---|---|---|---|---|---|---|---|---|---|---|---|---|
| TDH (clinical) | USA: TN | 346 | 0 | 0 | 0 | 0 | 41 | 18 | 6 | 17 | 0 | 305 | 305 | 0 |
| TDH (nonclinical) | USA: TN | 4 | 0 | 0 | 0 | 0 | 0 | 0 | 0 | 0 | 0 | 4 | 4 | 0 |
| Enterobase | USA | 27 | 1 | 1 | 0 | 0 | 13 | 5 | 3 | 5 | 0 | 13 | 13 | 0 |
| | CAN | 40 | 1 | 1 | 0 | 0 | 20 | 10 | 4 | 6 | 0 | 19 | 19 | 0 |
| | FRA | 67 | 4 | 2 | 2 | 0 | 38 | 10 | 20 | 8 | 0 | 25 | 23 | 2 |
| | GBR | 116 | 5 | 3 | 1 | 1 | 69 | 20 | 35 | 14 | 0 | 42 | 40 | 2 |
| | IRL | 5 | 1 | 1 | 0 | 0 | 2 | 0 | 1 | 1 | 0 | 2 | 2 | 0 |
| | LUX | 2 | 0 | 0 | 0 | 0 | 2 | 0 | 2 | 0 | 0 | 0 | 0 | 0 |
| | NLD | 5 | 1 | 1 | 0 | 0 | 3 | 0 | 2 | 1 | 0 | 1 | 1 | 0 |
| | NOR | 1 | 0 | 0 | 0 | 0 | 0 | 0 | 0 | 0 | 0 | 1 | 1 | 0 |
| | PRT | 2 | 0 | 0 | 0 | 0 | 0 | 0 | 0 | 0 | 0 | 2 | 2 | 0 |
| | TWN | 2 | 0 | 0 | 0 | 0 | 2 | 0 | 0 | 2 | 0 | 0 | 0 | 0 |
| | ZAF | 3 | 2 | 2 | 0 | 0 | 1 | 0 | 0 | 1 | 0 | 0 | 0 | 0 |
| RefSeq | USA | 26 | 1 | 0 | 0 | 0 | 19 | 17 | 0 | 2 | 0 | 6 | 6 | 0 |
| | VNM | 2 | 0 | 0 | 0 | 0 | 2 | 0 | 0 | 0 | 2 | 0 | 0 | 0 |
| | Unknown | 5 | 0 | 0 | 0 | 0 | 4 | 4 | 0 | 0 | 0 | 1 | 1 | 0 |
| Total | | 653 | 16 | 11 | 3 | 1 | 216 | 84 | 73 | 57 | 2 | 421 | 417 | 4 |

[a]TN, Tennessee; TDH, Tennessee Department of Health; USA, United States of America; CAN, Canada; FRA, France; GBR, Great Britian; IRL, Ireland; LUX, Luxembourg; NLD, Netherlands; NOR, Norway; PRT, Portugal; TWN, Taiwan; ZAF, South Africa; VNM, Vietnam.

(Fig. 1). From our Tennessee and global data sets, isolates belonged to eight different hierarchical clustering level 900 groups (HC900; i.e., with ≤900 core genome multilocus sequence typing [cgMLST] allelic differences) or cgMLST eBurst groups (ceBGs). ceBGs are equivalent to eBurst groups (eBGs) from legacy 7-gene MLST and correspond to serovar designations (20, 21). Typically, monophyletic serovars consist of a single eBG, while polyphyletic serovars consist of multiple eBGs (22–25). Lineage I isolates belonged to ceBGs 370, 12161, and 135213, lineage II isolates belonged to ceBGs 44, 391, and 401, and lineage III isolates belonged to ceBGs 120 and 65092 (Table 1, Fig. 1). All of these ceBGs have previously been associated with *S.* Newport except for 135213 (21). In our global phylogenetic analysis, most of the diversity at the HC100 level (≤100 cgMLST allelic differences) was found in lineage II (which contained 95 different HC100 clusters), followed by lineage III (74 HC100 clusters) (Table S1). Lineage I only contained 6 different HC100 clusters; this was likely a reflection of the rarity of this lineage compared to the two other lineages (10). The Tennessee *S.* Newport isolates all belonged to lineage II (n = 41) or lineage III (n = 309) (Table 1, Fig. 1). The low prevalence of Tennessee isolates belonging to lineage I was expected, as this lineage is known to be primarily isolated in Europe (10).

One limitation of this study was that the data set was not representative of the entire global diversity of this serovar. Most genomes downloaded from EnteroBase were isolated from humans in the United States and Europe; therefore, other countries and continents are underrepresented, particularly low- and middle-income countries (26–29), and the diversity of the data set is skewed toward countries that do extensive surveillance and sequencing of *Salmonella* clinical isolates and make these data publicly available. As genomic surveillance increases in underrepresented areas, a more complete picture of the diversity of clinical *S.* Newport will be achievable.

**The proportion of isolates that are outbreak related may be higher.** For routine surveillance, clusters are identified based on genomic distances, such as cgMLST allele differences, and then further investigated. From the lineage-specific phylogenetic analyses and using a threshold of 15 SNPs, followed by individual high-quality SNP (hqSNP) analyses using a threshold of 10 hqSNPs, we were able to retrospectively identify 18 potential epidemiological clusters consisting of a total of 84 isolates (Table 2), with 3 clusters in lineage II (Fig. 2) and 15 in lineage III (Fig. 3). The clusters contained an average of 4.67 clinical isolates (range, 3 to 12), with within-cluster average hqSNP distances ranging from 0 to 10.5

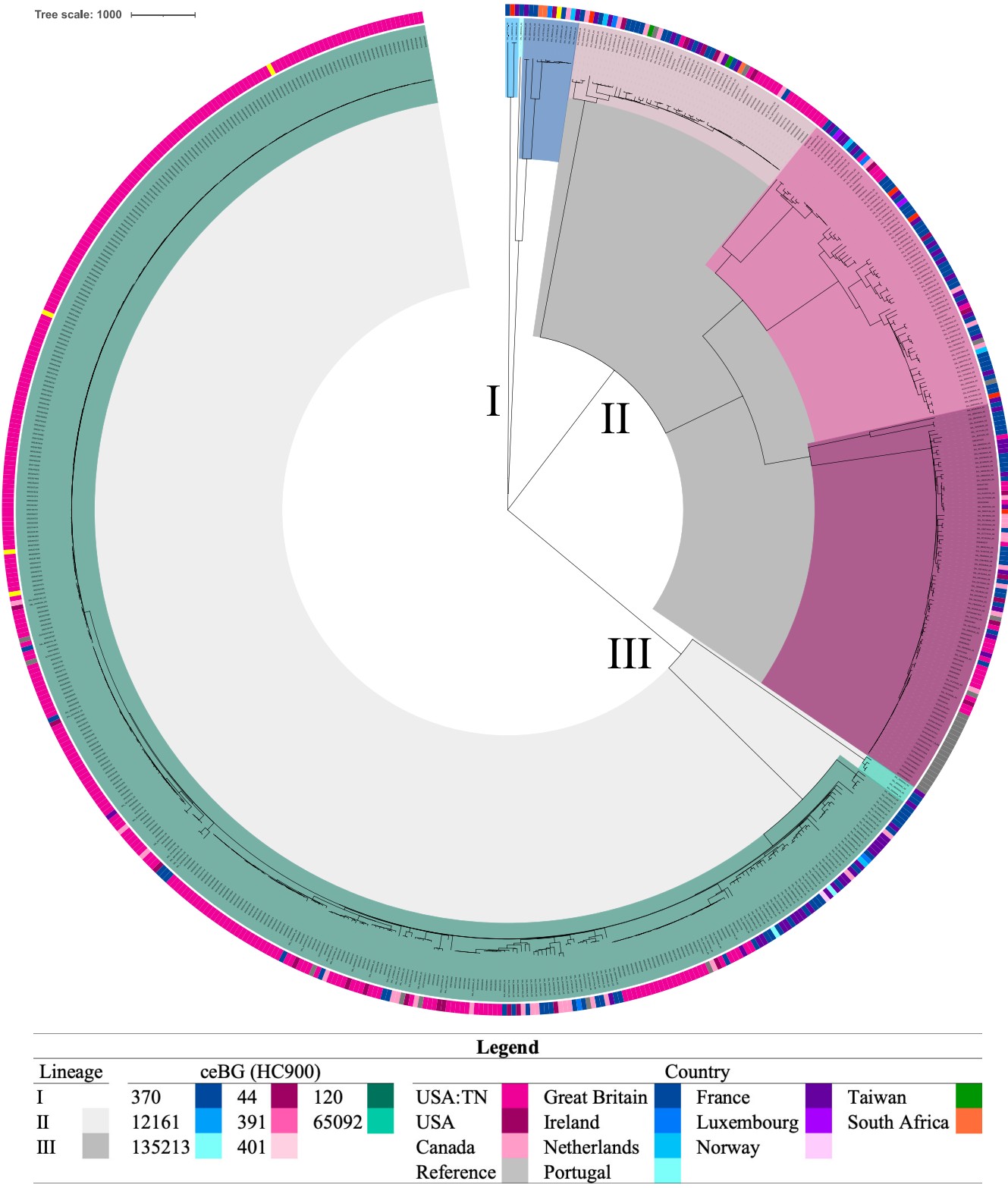

**FIG 1** Phylogeny of clinical *Salmonella* Newport isolates collected in Tennessee (2017 to 2021) and other global clinical isolates. The tree was constructed in MEGAX (43) using the neighbor-joining method (49). The tree is drawn to scale, with branch lengths representing the number of base differences at core SNP positions. The evolutionary distances were computed using the number of differences method (50). This analysis involved 654 isolates and 57,161 total core SNP positions. Lineage is indicated by clade shading: white, lineage I; dark gray, lineage II; light gray, lineage III. ceBG designations are indicated by clade shading (see the legend key). Country of isolation is indicated by colors in the outer ring (see legend key).

**TABLE 2** Potential cluster statistics

| Lineage | Cluster ID | No. of TN isolates | Avg hqSNP distance[a] (range) |
|---|---|---|---|
| 2 | 2-a | 12 | 6.56 (0–12) |
| | 2-d | 4 | 5.33 (1–9) |
| | 2-e | 4 | 4.00 (0–8) |
| 3 | 3-a | 4 | 4.67 (1–7) |
| | 3-b-2 | 4 | 5.17 (0–8) |
| | 3-c-2 | 4 | 0.50 (0–1) |
| | 3-e-2 | 9 | 10.22 (0–17) |
| | 3-e-4 | 3 | 0.67 (0–1) |
| | 3-i-2 | 3 | 2.00 (0–3) |
| | 3-i-6 | 3 | 8.67 (5–11) |
| | 3-j-12 | 3 | 8.00 (6–9) |
| | 3-j-2 | 4 | 5.50 (0–11) |
| | 3-j-3 | 6 | 8.60 (0–16) |
| | 3-j-4 | 3 | 0.00 |
| | 3-j-5-1 | 5 | 6.20 (0–10) |
| | 3-k | 6 | 0.00 |
| | 3-t | 3 | 4.67 (4–5) |
| | 3-v-2 | 4 | 10.50 (7–13) |

[a]The unit is hqSNPs.

hqSNPs. The proportion of TN clinical isolates that were part of potential outbreak clusters was 24%. Based on FoodNet data, it has been estimated that 5.9% of *Salmonella* cases are outbreak related (30, 31), with 3.0% for Tennessee, specifically. This may indicate that there are more related cases than previously thought, similar to what was found in *Campylobacter* spp. (32). However, isolation dates and other epidemiological data were not taken into account when identifying potential clusters in the current study, as would be done for cluster detection by public health departments. The current PulseNet guidelines for local cluster detection for *Salmonella* are as follows: three or more cases differing within 10 alleles by cgMLST, of which two differ within 5 alleles; isolation dates for all cases should be within the past 60 days of detection. Point source outbreaks and most continuous common source outbreaks would likely fall within that 60-day window. However, we chose to not include a time limit when identifying clusters in the current study, as we wanted to also be able to capture continuous common source potential outbreak clusters that spanned longer periods of time. Though these types of outbreaks may not make up a large portion of total illnesses, they are likely undetected if the isolation dates are not within the time window. Using genomic data for surveillance and analysis may help us to better understand factors that influence transmission, identify reservoirs, and detect and prevent outbreaks.

**Antibiotic resistance.** Drug-resistant nontyphoidal *Salmonella* is considered a serious threat in the United States (11). Therefore, it is important to monitor antibiotic resistance determinants in *Salmonella* clinical isolates. Antibiotics used to treat *Salmonella* infections include ampicillin, chloramphenicol, ciprofloxacin, ceftriaxone, trimethoprim-sulfamethoxazole, amoxicillin, and azithromycin (12–16). Among the Tennessee *S*. Newport isolates analyzed in this study, resistance to at least one antibiotic in the following classes was predicted: aminoglycoside (2.6%), tetracycline (4.3%), amphenicol (3.4%), folate pathway antagonist (3.1%), beta-lactam (2.9%), macrolide (2.3%), quinolone (2.3%), aminocyclitol (1.7%) (Table 3, Fig. 2 and 3). The predicted resistance to antibiotics used to treat *Salmonella* infections was at the following levels among Tennessee isolates: ampicillin resistance (2.9%), chloramphenicol (3.4%), ciprofloxacin (2.3%), ceftriaxone (0%), trimethoprim (2.3%), sulfamethoxazole (3.1%), amoxicillin (2.9%), and azithromycin (2.3%). Of the Tennessee isolates, 3.4% (*n* = 12) were predicted to be resistant to at least one essential antibiotic (ciprofloxacin, azithromycin, ceftriaxone, ampicillin, or trimethoprim-sulfamethoxazole), and 2.3% (*n* = 8) were predicted to be resistant to three or more.

The predicted resistance levels in Tennessee clinical *Salmonella* Newport isolates were lower (15 to 19 percentage points) than those reported by the National Antimicrobial

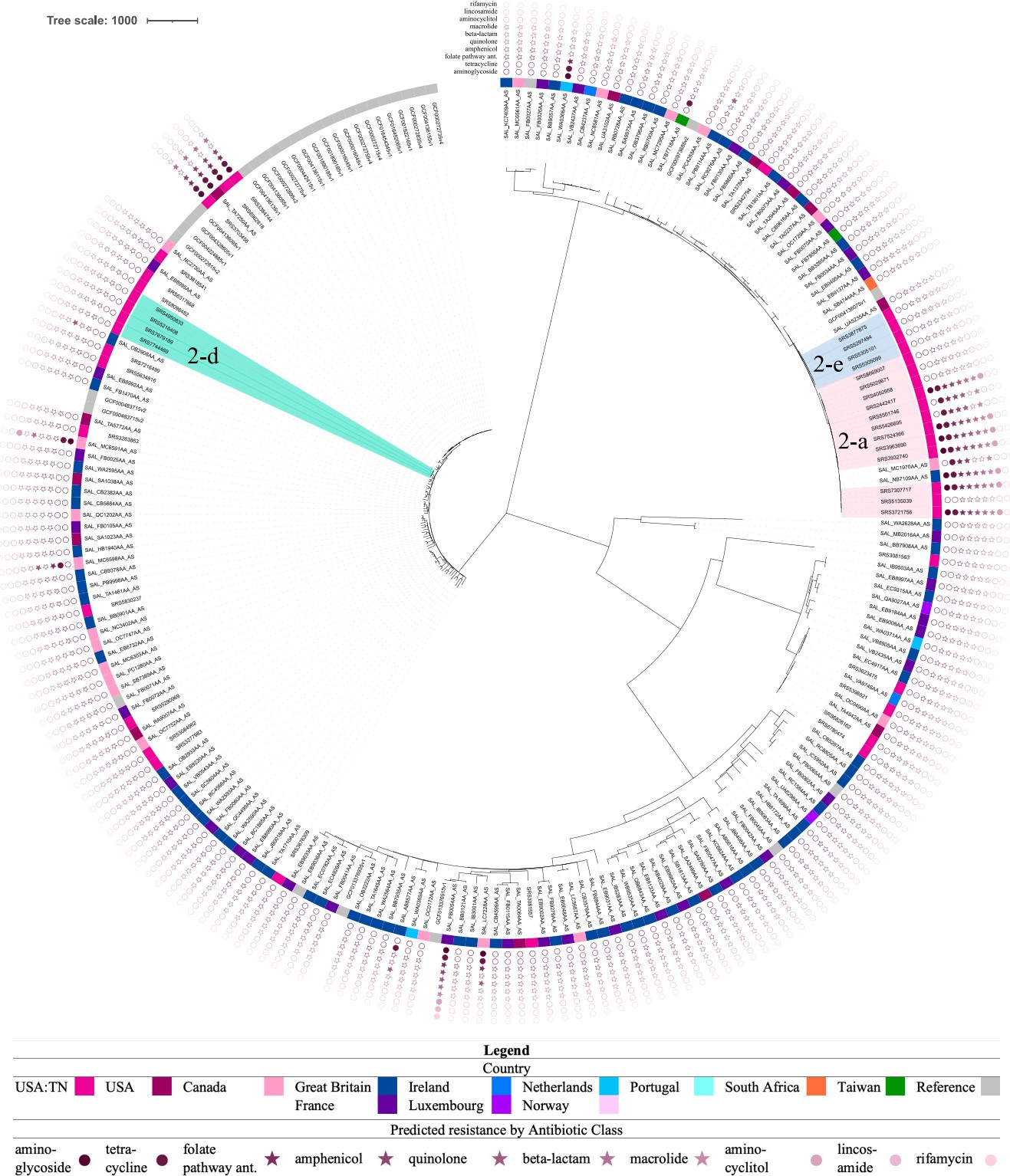

**FIG 2** Phylogeny of lineage II isolates, with potential clusters and antibiotic resistance determinants indicated. The tree was constructed in MEGAX (43) using the neighbor-joining method (49). The tree is drawn to scale, with branch lengths representing the number of base differences at core SNP positions. The evolutionary distances were computed using the number of differences method (50). This analysis involved 216 isolates and 37,118 total core SNP positions. Potential epidemiological clusters and their designations are indicated by clade shading: pink, 2-a; teal, 2-d; blue, 2-e. Country of isolation is indicated by colors in the inner ring (see legend key). Predicted resistance by antibiotic class is indicated by filled stars or circles in the outer ring set in the following order (from innermost to outermost): aminoglycoside, tetracycline, folate pathway antagonist, amphenicol, quinolone, beta-lactam, macrolide, aminocyclitol, lincosamide, and rifamycin. Stars indicate that a class contains an antibiotic(s) used to treat salmonellosis.

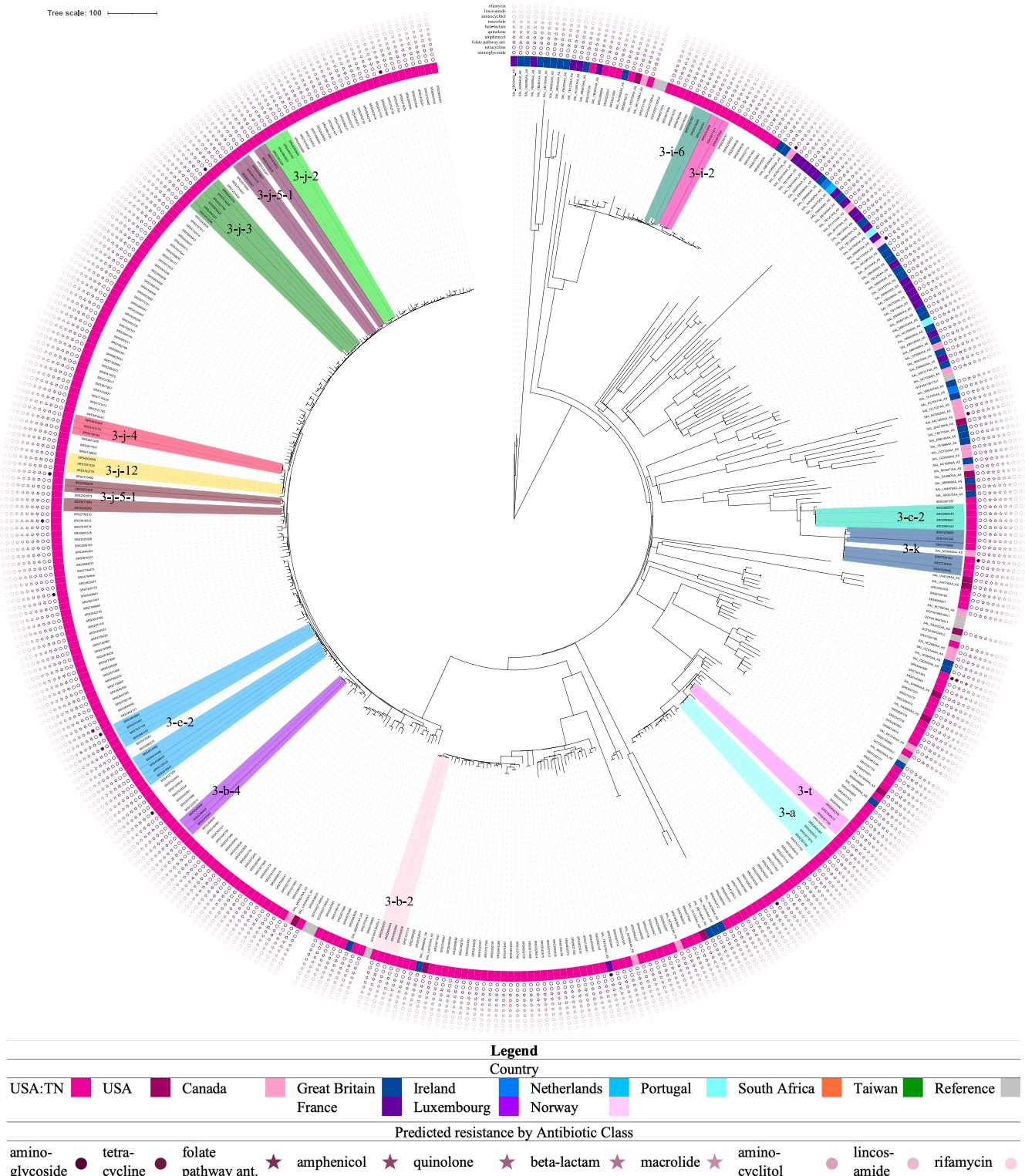

**FIG 3** Phylogeny of lineage III isolates, with potential clusters and antibiotic resistance determinants indicated. The tree was constructed in MEGAX (43) using the neighbor-joining method (49). The tree is drawn to scale, with branch lengths representing the number of base differences at core SNP positions. The evolutionary distances were computed using the number of differences method (50). This analysis involved 421 isolates and 28,321 total core SNP positions. Potential epidemiological clusters and their designations are indicated by clade shading (see legend key). Country of isolation is indicated by colors in the inner ring (see legend key). Predicted resistance by antibiotic class is indicated by filled stars or circles in the outer ring set in the following order (from innermost to outermost): aminoglycoside, tetracycline, folate pathway antagonist, amphenicol, quinolone, beta-lactam, macrolide, aminocyclitol, lincosamide, and rifamycin. Stars indicate that a class contains an antibiotic(s) used to treat salmonellosis.

**TABLE 3** Predicted antibiotic resistance by class, subclass, and antibiotic

| Class or subclass | Antibiotic[a] | TN isolates | | EnteroBase | | NARMS[c] | |
|---|---|---|---|---|---|---|---|
| | | No. | % | No. | % | No. | % |
| Aminocyclitol | | 6 | 1.71 | 3 | 1.11 | | |
| | Spectinomycin | 6 | 1.71 | 3 | 1.11 | | |
| Aminoglycoside | | 9 | 2.57 | 6 | 2.21 | 1,430 | 18.07 |
| | Apramycin | 0 | 0.00 | 2 | 0.74 | | |
| | Dibekacin | 0 | 0.00 | 2 | 0.74 | | |
| | Gentamicin | 0 | 0.00 | 2 | 0.74 | 33 | 0.42 |
| | Kanamycin | 0 | 0.00 | 1 | 0.37 | 62 | 0.78 |
| | Neomycin | 0 | 0.00 | 1 | 0.37 | | |
| | Netilmicin | 0 | 0.00 | 2 | 0.74 | | |
| | Sisomicin | 0 | 0.00 | 2 | 0.74 | | |
| | Streptomycin | 9 | 2.57 | 6 | 2.21 | 1,422 | 17.97 |
| | Tobramycin | 0 | 0.00 | 2 | 0.74 | | |
| Amphenicol | | 12 | 3.43 | 4 | 1.48 | 1,707 | 21.57 |
| | Chloramphenicol[b] | 12 | 3.43 | 4 | 1.48 | 1,707 | 21.57 |
| | Florfenicol | 12 | 3.43 | 4 | 1.48 | | |
| Beta-lactams | | 10 | 2.86 | 4 | 1.48 | | |
| Cephem | | | | | | 169 | 2.14 |
| | Cefepime | 0 | 0.00 | 1 | 0.37 | | |
| | Cefotaxime | 4 | 1.14 | 2 | 0.74 | | |
| | Cefoxitin | 4 | 1.14 | 1 | 0.37 | 158 | 2.00 |
| | Ceftazidime | 4 | 1.14 | 2 | 0.74 | | |
| | Ceftiofur | 0 | 0.00 | 0 | 0.00 | 168 | 2.12 |
| | Ceftriaxone[b] | 0 | 0.00 | 1 | 0.37 | 168 | 2.12 |
| | Cephalothin | 0 | 0.00 | 2 | 0.74 | | |
| | Cephalotin | 0 | 0.00 | 1 | 0.37 | | |
| Monobactam | Aztreonam | 0 | 0.00 | 1 | 0.37 | | |
| Penicillin | | | | | | 1,453 | 18.36 |
| | Amoxicillin[b] | 10 | 2.86 | 4 | 1.48 | | |
| | Ampicillin[b] | 10 | 2.86 | 4 | 1.48 | 1,453 | 18.36 |
| | Piperacillin | 10 | 2.86 | 4 | 1.48 | | |
| | Ticarcillin | 4 | 1.14 | 3 | 1.11 | | |
| Beta-lactam combination agent | | | | | | 159 | 2.01 |
| | Amoxicillin + clavulanic acid | 4 | 1.14 | 1 | 0.37 | 159 | 2.01 |
| | Piperacillin + tazobactam | 4 | 1.14 | 1 | 0.37 | | |
| | Ticarcillin + clavulanic acid | 4 | 1.14 | 1 | 0.37 | | |
| | Ampicillin + clavulanic acid | 4 | 1.14 | 1 | 0.37 | | |
| Unknown beta-lactam | | 0 | 0.00 | 1 | 0.37 | | |
| Folate pathway antagonist | | 11 | 3.14 | 7 | 2.58 | 1,748 | 22.09 |
| | Sulfamethoxazole[b] | 11 | 3.14 | 7 | 2.58 | 1,748 | 22.09 |
| | Trimethoprim[b] | 8 | 2.29 | 5 | 1.85 | | |
| | Trimethoprim-sulfamethoxazole[b] | | | | | 1,592 | 20.12 |
| Lincosamide | | 0 | 0.00 | 1 | 0.37 | | |
| | Lincomycin | 0 | 0.00 | 1 | 0.37 | | |
| Lipopeptide | | | | | | 1 | 0.01 |
| Macrolide | | 8 | 2.29 | 3 | 1.11 | 1,569 | 19.83 |
| | Azithromycin[b] | 8 | 2.29 | 3 | 1.11 | 1,569 | 19.83 |
| | Erythromycin | 8 | 2.29 | 3 | 1.11 | | |
| | Spiramycin | 8 | 2.29 | 3 | 1.11 | | |
| | Telithromycin | 8 | 2.29 | 3 | 1.11 | | |
| Polymyxin | Colistin | 0 | 0.00 | 0 | 0.00 | 1 | 0.01 |
| Quinolone | | 8 | 2.29 | 8 | 2.95 | 1,370 | 17.31 |
| | Ciprofloxacin[b] | 8 | 2.29 | 8 | 2.95 | 1,370 | 17.31 |
| | Nalidixic acid | 0 | 0.00 | 1 | 0.37 | 16 | 0.20 |
| Rifamycin | Rifampin | 0 | 0.00 | 1 | 0.37 | | |
| Tetracycline | | 15 | 4.29 | 10 | 3.69 | 1,806 | 22.82 |
| | Doxycycline | 15 | 4.29 | 10 | 3.69 | | |
| | Minocycline | 3 | 0.86 | 1 | 0.37 | | |
| | Tetracycline | 15 | 4.29 | 10 | 3.69 | 1,806 | 22.82 |

[a]Antibiotics for which no resistance was predicted in any of the study isolates and for which NARMS data were not listed.
[b]An antibiotic used to treat *Salmonella* infections.
[c]Data from NARMS Now human data (https://wwwn.cdc.gov/narmsnow/), included for comparison purposes.

Resistance Monitoring System (NARMS) for U.S. clinical cases (33) for the following antibiotics: sulfamethoxazole, trimethoprim, tetracycline, chloramphenicol, azithromycin, ampicillin, streptomycin, and ciprofloxacin (Table 3). This discrepancy could have been due to differences between the ResFinder and NARMS databases used for predicting antibiotic resistance. Future work should be conducted to determine if these differences are due to database inconsistencies or if Tennessee does have lower rates of AR than other regions of the United States. If there are database inconsistencies, these should be rectified to ensure that genotypic AR predictions are as concordant as possible with AR phenotypes. All other predicted resistance levels were within 0 to 2.2% of the NARMS human predictions for antibiotics for which data were available (Table 3).

One of the clusters that was identified in lineage II (cluster 2-a) contained several isolates that were predicted to be multidrug resistant. They were predicted to be resistant to antibiotics in the aminoglycoside, tetracycline, folate pathway antagonist, amphenicol, quinolone, beta-lactam, macrolide, and/or aminocyclitol classes.

**Conclusions.** *Salmonella* Newport is a clinically important *Salmonella* serovar that causes a high number of illnesses. Evaluating the relatedness of clinical isolates can provide guidance for phylogenetic analyses and cluster detection for public health surveillance and response. In this study, we demonstrated the population structure of *S.* Newport isolates from Tennessee in the context of the overall global diversity, and we found that all Tennessee isolates belonged to Newport phylogenetic lineages II and III. Additionally, we identified a greater proportion of Tennessee isolates belonged to putative epidemiological clusters than the proportion of isolates of this serovar that were outbreak related. This study provided insights on *Salmonella enterica* serovar Newport that can impact public health surveillance and response and serve as a foundational context for the *Salmonella* research community.

## MATERIALS AND METHODS

IDs and metadata were provided by the Tennessee Department of Health for clinical *S.* Newport isolates sequenced by the state public health laboratory as part of routine surveillance, with collection dates from 2017 to 2021. Raw Illumina reads were downloaded from the NCBI Sequence Read Archive (SRA) using fastq-dump, and Trimmomatic (34) was used to filter poor-quality reads and trim poor-quality bases. FastQC v0.11.9 (35) was used to check the quality of read files, and MultiQC v1.11 (36) was used to compile the outputs and generate a report. Trimmed reads were assembled using SPAdes v3.13.1 (37), and resulting contigs of <l kb or with <5× average read coverage were filtered out. Assembly statistics were generated using QUAST (38), MultiQC (36), and a custom script to determine average coverage. Assemblies were removed if they did not meet all of the following inclusion criteria: total length of 4.5 to 5.1 Mb, ≤200 contigs, ≥10× average read coverage, and 50 to 53% G+C content.

**Global phylogenetic analysis.** The *Salmonella* database (23) on EnteroBase (20) was queried for isolates with "human" listed as the source niche in the strain metadata and "Newport" listed as the serovar in the strain metadata or experimental data (SISTR1 [39] or SeqSero 2 [40]). GrapeTree (41) was used to create a cgMLST + HierCC (core-genome multilocus sequence typing + hierarchical clustering of cgMLST) minimal spanning tree using a RapidNJ algorithm on EnteroBase with all of the resulting strains. The initial global data set consisted of 2,416 isolates. A representative portion of the data set was employed by selecting a single isolate from each country from each HC100-level cluster (≤100 cgMLST allelic differences). If more than one isolate was available for each country and state, the representative strain was chosen based on assembly quality metrics. The representative global data set consisted of 270 isolates, and those assemblies were downloaded from EnteroBase. Additionally, complete genome assemblies (*n* = 40) were downloaded from NCBI RefSeq to serve as references. The entire data set used for phylogenetic analysis included 653 isolates: 350 from Tennessee, 270 from Enterobase, and 33 from RefSeq. Phylogeny was determined using kSNP3 (42), a reference-free method for SNP detection, and the resulting core SNP fasta file was used to create a core SNP matrix and an unrooted neighbor-joining tree in MegaX (43). The distances were computed using the number of differences method, with uniform substitution rates, and 100 bootstrap (44) replications were conducted. Based on the phylogeny, the isolates were separated by lineage, and KSNP3 was again used to evaluate the phylogenies of lineages II and II individually.

**Putative cluster detection.** Putative clusters were identified as previously described (32). Candidate clusters of *Salmonella* Newport consisting of at least three isolates were identified using SNP distances from the lineage-specific phylogenetic analyses and a threshold of 15 SNPs. This threshold of 15 was chosen to be more inclusive than the 10-allele differences specified in the PulseNet guidelines. Each individual cluster candidate was then further evaluated by a higher-resolution hqSNP analysis with the CFSAN snp-pipeline (v2.2.1) (45), using one of the included isolate assemblies as the reference; a threshold of 10 hqSNPs was used to confirm each as a putative cluster.

**Antibiotic resistance determinants.** Antibiotic resistance determinants (acquired resistance genes and chromosomal point mutations mediating resistance) were identified from the *Salmonella* Newport assemblies using ResFinder/PointFinder (version 4.2.0; ResFinder database 2022-05-24; PointFinder database 2022-04-

22), with an identity threshold of 0.8 and minimum coverage of 0.6 (46). The original prediction for amikacin resistance was 100%, due to all isolates containing the *aac(6')-Iaa* gene. However, this gene has been shown to be cryptic and no longer confer phenotypic resistance to aminoglycosides (25, 47, 48), so the results were adjusted accordingly. The NARMS human clinical ABR data were downloaded from NARMS Now (33) for comparison purposes.

**Data availability.** All Tennessee isolate sequencing reads are available from the NCBI SRA, and SRA IDs are provided in Data Set S1 in the supplemental material. EnteroBase genomes are available from EnteroBase, and the relevant IDs are provided in Data Set S1.

## SUPPLEMENTAL MATERIAL

Supplemental material is available online only.

**SUPPLEMENTAL FILE 1**, XLSX file, 0.1 MB.

## ACKNOWLEDGMENTS

This work was supported by Epidemiology and Laboratory Capacity for Infectious Diseases (grant number NU50CK000528). We thank the Sequencing Department of the Division of Laboratory Services, Tennessee Department of Health, for their assistance with DNA sequencing as part of routine surveillance and Christina Moore for leading the Sequencing Team.

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
