## [Reviewer comments · Microbiology Spectrum]

Microbiology Spectrum

Phylogeny and Genomic Characterization of Clinical *Salmonella enterica* serovar Newport Collected in Tennessee

Lauren Hudson, William Andershock, Xiaorong Qian, Paula Gibbs, Kelly Orejuela, Katie Garman, John Dunn, and Thomas Denes

Corresponding Author(s): Thomas Denes, The University of Tennessee Knoxville

Review Timeline:

Submission Date:	September 26, 2022
Editorial Decision:	November 14, 2022
Revision Received:	November 17, 2022
Accepted:	November 21, 2022

Editor: Sandeep Tamber

Reviewer(s): The reviewers have opted to remain anonymous.

Transaction Report:

DOI: <https://doi.org/10.1128/spectrum.03876-22>

November 14, 2022

Dr. Thomas G Denes
The University of Tennessee Knoxville
Food Science
2600 River Drive
Knoxville 37996-3591

Re: Spectrum03876-22 (Phylogeny and Genomic Characterization of Clinical *Salmonella enterica* serovar Newport Collected in Tennessee)

Dear Dr. Thomas G Denes:

Link Not Available

Sincerely,

Sandeep Tamber

Journals Department
Editor comments:

Both reviewers expressed major reservations about moving forward with this submission. There were concerns about missing information and the preliminary nature of the results and interpretation. If you decide to submit a revision, please ensure all reviewer comments are addressed, and the significance of the the findings are clearly laid out. A minor consideration is the use of the phrase "TN isolates" may be confusing since Tennessee is also the name of a *Salmonella* serovar.

Reviewer comments:

Reviewer #1 (Comments for the Author):

Hudson et al. presents their phylogenetic analyses of *Salmonella* Newport isolates obtained by the Tennessee Department of

Health. The diversity and relationships of the Tennessee isolates were also analyzed in the context of the global population. Overall, there were major parts that were missing in the manuscript that greatly affected my assessment of the study.

Major comments:

1. Importance: This section needs major revision. As it is written, it is merely a list of the objectives of the study.
2. M&M, lines 117-119: The methods for phylogenetic reconstruction is insufficiently described. Need to state the kind of tree (maximum likelihood, maximum parsimony, neighbor joining, Bayesian, etc.), length of alignment, alignment method used, rooting method used, nucleotide substitution model used (including gamma distribution), whether bootstrapping was done and if yes, how many bootstrap replicates
3. M&M, lines 122-127: This is confusing. What was the rationale for choosing 15 SNPs as threshold to define a cluster? What was the rationale for doing a second clustering with a different SNP threshold?
4. Results: Since the study (from the title) is about Salmonella from Tennessee, I am surprised there was no findings presented about Salmonella from Tennessee only. The entire manuscript describes the global population structure. For clarity, additional analyses, figure and section in the results describing only the Tennessee samples should have been presented before the global population structure was presented.
5. The Discussion component of the combined Research and Discussion was missing. The implications of the findings were not described.
6. Table 2, Abstract line 27, Results line 176: How the 18 epidemiological clusters among the Tennessee isolates was identified was not explained. Without a phylogenetic tree of the Tennessee isolates, this is difficult to assess.
7. Conclusions "These types of studies provide useful information about the relationship between phylogeny and strain characteristics, which may be useful for making predictions about isolates, such as if they are likely outbreak-related or multi-drug resistant." This statement is vague and the study does not discuss how this can be used as a predictive tool. What the authors meant by relationship between phylogeny and strain characteristics was unclear.

Minor comments:

1. Abstract: Include the number of genomes analyzed
2. Abstract, lines 20-23: Sentence is too long. I suggest that it is split into two shorter sentences.
3. Intro, line 52 and throughout the manuscript: Use Tennessee not TN
4. Intro, line 62 "S. Newport is and is recognized": Remove "and is"
5. Intro, line 65: W in wild should be lowercase
6. Intro, line 66 "resistance during food processing": resistance to what?
7. Intro, lines 71, 77: lineages not linages
8. Intro, line 91: Spell out ABR the first time it is used.
9. M&M: How many genomes from Tennessee were used in this study?
10. M&M, line 95: laboratory not lab
11. M&M, line 101: statistics not statics
12. M&M, line 103: did not (do not use contraction didn't)
13. M&M, line 108: GrapeTree not Grape Tree. Also include citation for GrapeTree
14. M&M, line 108: spell out cgMLST + Hier CC
15. M&M, line 118: spell out SNP the first time it is used
16. There were too many acronyms throughout the manuscript that were not defined: MLST, hqSNP
17. M&M, lines 130-131: version of ResFinder and PointFinder used? Database used?
18. M&M, lines 135: What is NARMS Now?
19. M&M, line 137: accession numbers, not accessions
20. Results, line 140: You already spelled out Tennessee Department of Health (TDH) in the M&M. No need to do it again in this section.
21. Figure 2: I don't understand why the circles that form the outer rings are fading from inner to outer rings?
22. All phylogenetic trees: The tip labels are too small to be legible, and thus uninformative. Better to remove them or increase the font size.

Reviewer #2 (Comments for the Author):

This study analyzed by genomic tools 350 clinical strains of Salmonella Newport isolated in Tennessee from 2017 to 2021 and compared them to strains isolated in several other countries.

The authors determined the lineages to which these strains belong, and noted the presence of 18 potential clusters not routinely detected, which highly frequent in several surveillance programs worldwide. They also used genomic tools for the prediction of antibiotic resistance. This study is well constructed and quite complete but does not raise any scientific findings worthy of a scientific article in this journal. With the generalization of genomics for Salmonella surveillance, this can be published as a comprehensive report of Salmonella Newport surveillance in Tennessee, to inform epidemiologists and microbiologists in the

region.

Staff Comments:

Preparing Revision Guidelines

Please return the manuscript within 60 days; if you cannot complete the modification within this time period, please contact me. If you do not wish to modify the manuscript and prefer to submit it to another journal, please notify me of your decision immediately so that the manuscript may be formally withdrawn from consideration by Microbiology Spectrum.

We thank the reviewers for the constructive feedback and have provided responses below to all comments in blue font (line numbers correspond to those shown in the marked-up version of the manuscript).

Editor comments:

Both reviewers expressed major reservations about moving forward with this submission. There were concerns about missing information and the preliminary nature of the results and interpretation. If you decide to submit a revision, please ensure all reviewer comments are addressed, and the significance of the findings are clearly laid out. A minor consideration is the use of the phrase "TN isolates" may be confusing since Tennessee is also the name of a Salmonella serovar.

We believe we have fully addressed the reviewer comments below and are confident this manuscript is technically sound, complete, and that it is useful to the community. Our understanding of the *Microbiology Spectrum* scope leads us to believe our manuscript is a suitable fit for publication in this journal (see response to reviewer 2). Thank you for pointing out the potential for confusion with serovar Tennessee. To address this, we have changed the wording to refer to "isolates from Tennessee (TN)" throughout. Additionally, we have moved the Materials and Methods section to the end to follow the ASM convention.

Reviewer comments:

Reviewer #1 (Comments for the Author):

Hudson et al. presents their phylogenetic analyses of Salmonella Newport isolates obtained by the Tennessee Department of Health. The diversity and relationships of the Tennessee isolates were also analyzed in the context of the global population. Overall, there were major parts that were missing in the manuscript that greatly affected my assessment of the study.

Major comments:

1. Importance: This section needs major revision. As it is written, it is merely a list of the objectives of the study.

The objectives of the study (found in introduction lines 98-104) are not the same or similar to what we included in the importance section. However, we do think the importance we submitted was too general, and thus have revised it (lines 37-45).

The journal guidelines (found here) for this section specify that it "should provide a nontechnical explanation of the significance of the study to the field." We believe our revised importance abstract meets this criterion.

2. M&M, lines 117-119: The methods for phylogenetic reconstruction is insufficiently described. Need to state the kind of tree (maximum likelihood, maximum parsimony,

neighbor joining, Bayesian, etc.), length of alignment, alignment method used, rooting method used, nucleotide substitution model used (including gamma distribution), whether bootstrapping was done and if yes, how many bootstrap replicates

The kind of tree (neighbor-joining) was already stated in the methods (line 233) and in each of the figure legends (lines 547, 556, and 569). The phylogeny was determined based on core SNPs, not genome alignments, so that information is not relevant – however, the number of SNP positions that were used are listed in each of the figure legends (lines 550, 559, and 572). Readers who may not be familiar with kSNP can follow the included reference (“kSNP3.0: SNP detection and phylogenetic analysis of genomes without genome alignment or reference genome”). We added that this is an unrooted tree (line 233) (Note: including outgroups with this analysis method would greatly reduce the number of core SNPs that can be compared across genomes). We added the information about the model and bootstrapping (“The distances were computed using the number of differences method, with uniform substitution rates, and 100 bootstrap replications were conducted.” [lines 234-235]).

3. M&M, lines 122-127: This is confusing. What was the rationale for choosing 15 SNPs as threshold to define a cluster? What was the rationale for doing a second clustering with a different SNP threshold?

The thresholds are based on the current PulseNet guidelines for local cluster detection (described in lines 154-157). The first round of cluster identification was done with the kSNP SNP distances from the lineage-specific phylogenetic analyses. We used a slightly higher threshold (15 instead of 10), so that it would be more inclusive. Then for confirmation, each individual cluster was evaluated by using the CFSAN snp-pipeline – then cluster inclusion was based on hqSNP distances and a threshold of 10 was used as this would be similar to the PulseNet guidelines. This section was revised for clarity and to add context (lines 241-245).

4. Results: Since the study (from the title) is about Salmonella from Tennessee, I am surprised there was no findings presented about Salmonella from Tennessee only. The entire manuscript describes the global population structure. For clarity, additional analyses, figure and section in the results describing only the Tennessee samples should have been presented before the global population structure was presented.

We disagree with this statement (that “no findings presented about Salmonella from Tennessee only”). First, our cluster analysis and prediction of the potential proportion of isolates associated with outbreaks were exclusively conducted on the TN isolates. Second, the analysis that included other isolates included them as context/controls to better understand the TN isolates (we could not draw conclusions about the diversity of the TN strains unless we compared that to the global diversity). We believe an additional analysis showing diversity of only TN samples would be redundant and lacking appropriate context.

5. The Discussion component of the combined Research and Discussion was missing. The implications of the findings were not described.

We have highlighted 744 words in teal in the marked-up version of the manuscript portions in the Results and Discussion section that we believe qualify as Discussion, including the Conclusions subheading, which falls under this section. We did find that there was no discussion of the genomic assembly statistics and have added the following sentence of discussion to this section: "The assembly lengths and G+C content were within expected ranges for this serovar" (lines 109-110).

6. Table 2, Abstract line 27, Results line 176: How the 18 epidemiological clusters among the Tennessee isolates was identified was not explained. Without a phylogenetic tree of the Tennessee isolates, this is difficult to assess.

The identification of the putative clusters was described in the Methods section (lines 238-245) and again briefly in the Results and Discussion section (lines 143-144). Additionally, the putative clusters are visually indicated (colored shading and labeled) in the phylogenetic trees in Figures 2 and 3.

7. Conclusions "These types of studies provide useful information about the relationship between phylogeny and strain characteristics, which may be useful for making predictions about isolates, such as if they are likely outbreak-related or multi-drug resistant." This statement is vague and the study does not discuss how this can be used as a predictive tool. What the authors meant by relationship between phylogeny and strain characteristics was unclear.

We agree. We have removed this sentence and replaced it with this: "This study provides insight on *Salmonella* serovar Newport that can impact public health surveillance and response and serves as foundational context for the *Salmonella* research community." (lines 203-205). We believe this is appropriate for the concluding sentence of the manuscript.

Minor comments:

8. Abstract: Include the number of genomes analyzed
This was added (lines 24-25).

9. Abstract, lines 20-23: Sentence is too long. I suggest that it is split into two shorter sentences.

This sentence was split into two sentences and now reads "*Salmonella enterica* subsp. *Enterica* serovar Newport (*S. Newport*) is a clinically and epidemiologically significant serovar in the United States. It is the second most prevalent clinically isolated *Salmonella* serovar in the U.S. and can contaminate a wide variety of food products." (lines 20-23).

10. Intro, line 52 and throughout the manuscript: Use Tennessee not TN

We think that using the abbreviation, which we define in lines 28 and 61, may help in preventing confusion with *Salmonella* serovar Tennessee (see editor comments above).

11. Intro, line 62 “S. Newport is and is recognized”: Remove “and is”
This was removed (line 71).
12. Intro, line 65: W in wild should be lowercase
This was corrected (line 74).
13. Intro, line 66 “resistance during food processing”: resistance to what?
This sentence was revised and now reads “Additionally, *S. Newport* can invade plant tissues, which renders washing and disinfection ineffective for eliminating this pathogen during processing” (lines 75-77).
14. Intro, lines 71, 77: lineages not linages
This was corrected in both places (lines 82 and 86)
15. Intro, line 91: Spell out ABR the first time it is used.
This was added (line 103).
16. M&M: How many genomes from Tennessee were used in this study?
This was on lines 106 and 230.
17. M&M, line 95: laboratory not lab
This was changed (line 208).
18. M&M, line 101: statistics not statics
This was corrected (line 214).
19. M&M, line 103: did not (do not use contraction didn't)
This was corrected (line 215).
20. M&M, line 108: GrapeTree not Grape Tree. Also include citation for GrapeTree
This was corrected and citation was added (line 221).
21. M&M, line 108: spell out cgMLST + Hier CC
This was added (lines 221-222)
22. M&M, line 118: spell out SNP the first time it is used
This was added (lines 232).
23. There were too many acronyms throughout the manuscript that were not defined:
MLST, hqSNP
These were spelled out (lines 243 and 120-121).
24. M&M, lines 130-131: version of ResFinder and PointFinder used? Database used?

This information was added (lines 248-249).

25. M&M, lines 135: What is NARMS Now?

We added the full name for NARMS (line 253). NARMS Now is a tool available on the CDC website that can be used to view and/or download AMR data. There is a citation in the text (reference 33) that readers can use to access or learn more about the tool.

26. M&M, line 137: accession numbers, not accessions

Accession was changed to "IDs" (lines 256-257).

27. Results, line 140: You already spelled out Tennessee Department of Health (TDH) in the M&M. No need to do it again in this section.

We have moved the Materials and Methods section to the end to follow the ASM convention. So we left it spelled out here (line 107) and just used the acronym in the Materials and Methods (line 207).

28. Figure 2: I don't understand why the circles that form the outer rings are fading from inner to outer rings?

The circles indicating predicted ABR are different shades of the same color, the darkest for the innermost class and the lightest for the outermost class (see legends at the bottom of the figures). While creating our figures, we kept accessibility in mind. Using varying shades of the same color is one way to display information in an accessible way.

29. All phylogenetic trees: The tip labels are too small to be legible, and thus uninformative. Better to remove them or increase the font size.

The tip labels aren't necessary for viewing and interpreting the trees. However, we did leave them in the figures, so that if a reader did want to know isolate IDs, they could still be found. We believe this may help support any research that may build on these findings. This article will be published online and the figures are in a format so they can be zoomed in enough to clearly read the tip labels (isolate IDs).

Reviewer #2 (Comments for the Author):

This study analyzed by genomic tools 350 clinical strains of Salmonella Newport isolated in Tennessee from 2017 to 2021 and compared them to strains isolated in several other countries. The authors determined the lineages to which these strains belong, and noted the presence of 18 potential clusters not routinely detected, which highly frequent in several surveillance programs worldwide. They also used genomic tools for the prediction of antibiotic resistance. This study is well constructed and quite complete but does not raise any scientific findings worthy of a scientific article in this journal. With the generalization of genomics for Salmonella

surveillance, this can be published as a comprehensive report of Salmonella Newport surveillance in Tennessee, to inform epidemiologists and microbiologists in the region.

We believe that our manuscript that is “well constructed and quite complete” by definition should meet the criteria for publishing in *Microbiology Spectrum*. We would agree with the assessment that it’s “not worthy of a scientific article in this journal” if we had submitted to *Applied and Environmental Microbiology* or *Journal of Clinical Microbiology*; however, we very intentionally submitted to *Microbiology Spectrum* as we believe this is the ideal venue for incremental yet valuable contributions to the field. Below we will share the scope as we believe understanding this is critical for fairly evaluating submissions to this journal.

The scope of this journal (found here) states “Rather than making subjective evaluations of potential impact, *Microbiology Spectrum* publishes research studies that are of high technical quality and are useful to the community”

November 21, 2022

Dr. Thomas G Denes
The University of Tennessee Knoxville
Food Science
2600 River Drive
Knoxville 37996-3591

Re: Spectrum03876-22R1 (Phylogeny and Genomic Characterization of Clinical *Salmonella enterica* serovar Newport Collected in Tennessee)

Dear Dr. Thomas G Denes:

Your manuscript has been accepted, and I am forwarding it to the ASM Journals Department for publication. You will be notified when your proofs are ready to be viewed.

Sincerely,

Sandeep Tamber
Editor, Microbiology Spectrum

Journals Department
Supplemental Dataset 1: Accept